# Psychology at the screen: Investigating the current VAR protocol

Daniel Walker[1]*, Cara E. Staniforth[1], Jordan Thomas[2], Fallon Parker[1], Usman Khizar[1], Nathaniel J. Lawton[1], Tamryn Vermaak[3], Jade L. Jukes[1]

1 University of Bradford, Bradford, United Kingdom, 2 NHS Talking Therapies Liverpool, Liverpool, United Kingdom, 3 Manchester Metropolitan University, Manchester, United Kingdom

* d.walker5@bradford.ac.uk

## Abstract

This study aimed to investigate the psychological process of the current Video Assistant Referee (VAR) protocol by calculating, i) the prevalence of overturn/maintain decisions by the referee at the pitch-side monitor, ii) the likelihood that the referee would overturn their decision based on external factors, iii) the likelihood that the VAR would intervene based on external factors. 1520 matches were reviewed across 2021/22–2024/25 seasons in the English Premier League where the referee was advised to consult the pitch-side monitor 250 times, overturning their original decision 95% of the time. Binary Logistic Regression found that external factors did not predict the referee's decision or VAR intervention. The high prevalence of overturn decisions may be expected as referees are being advised to review potential errors and interestingly maintain decisions were more prevalent when related to the home team, potentially due to proximity bias of the home crowd.

## Introduction

The English Premier League introduced the Video Assistant Referee (VAR) at the beginning of the 2019/2020 season after clubs voted unanimously in favour of it in November 2018. Since then, controversy still arises [1–3] but the Premier League defend that the introduction of VAR has facilitated football and that teams are in favour of keeping it [4]. There is also evidence that the number of correct decisions has increased across nations since its introduction [5]. However, VAR has its critics over the negative impact on the matchday fan such as fans "being kept in the dark", it "ruining the moment" and the number of stoppages it causes to a game [6,7]. While spectators and pundits often voice their views, the effect of the VAR protocol on referee decision making is still uncertain. It is crucial for referees to make accurate and timely decisions [8] under pressure where errors can have significant consequences [9]. Therefore, it is important to understand the external factors that may influence decision making during the current VAR protocol.

**Data availability statement:** All data is freely available at https://osf.io/zjrp7/overview.

**Funding:** The author(s) received no specific funding for this work.

**Competing interests:** The authors have declared that no competing interests exist.

The primary roles of the VAR, and Assistant VAR (AVAR) are to support the on-field referee when a potential error has occurred. They are described as video match officials (VMOs) and assist the referee in accordance with the Laws of the Game [10] and the VAR protocol [11] which are both set by the International Football Association Board (IFAB). The VAR can assist the referee using replay footage for a "clear and obvious error" or a "serious missed incident" relating to a goal/no goal, penalty/no penalty, direct red card (not a second caution) or a case of mistaken identity [12]. The AVAR supports the VAR by observing the television footage while the VAR is occupied with a check or a review of another incident as well as other communication responsibilities [12]. The introduction of the VMOs has therefore influenced the role of the referee in that they may be advised to review one of their decisions at the pitch-side monitor by the VAR or AVAR.

Naturally, when it comes to decision making, particularly in sport, the accuracy is of specific interest to many. Spitz et al. [5] are among those that have attempted to investigate the impact that the introduction of VMOs have had on the accuracy of decision making. This group of researchers reviewed 2195 competitive football matches across thirteen countries (Australia, Belgium, China, Czechia, England, France, Germany, Italy, The Netherlands, Poland, Portugal, South Korea, and USA) over the 2016/17 and 2017/18 seasons. It was found that the VAR conducted 9732 checks for potential match-changing incidents (goals, penalty decisions, direct red cards, and mistaken identity) with this leading to 795 on-field reviews at the monitor. Accuracy of correct decisions appeared to increase to 98.3% from the initial 92.1% prior to the VAR introduction. Given football associations and governing bodies are interested in the accuracy of decision making by officials, work like Spitz et al.'s [5] acts as justification for the introduction of VAR. However, as this work reviewed data in matches between 2016 and 2018, the findings could be considered outdated due to the ever-changing and volatile environment of football and the VAR protocol.

The present study was less interested in the accuracy of referee decisions, but instead the psychology of the on-field referee as they review their decision at the pitch-side monitor. While Spitz et al. [5] provide evidence that the VAR process leads to improved decision accuracy, this could still be questioned. Spitz et al. [5] admit themselves that the laws of the game leave room for interpretation at times, and while they took steps to mitigate this by omitting "grey zone" decisions, this may still be problematic. Removing decisions where not all referees agreed on the "correct decision" was a sensible approach, but when this is with 5–10 referees, it is conceivable that their conclusions could be considered contentious. This example may be seen in practice where six officials come to the same conclusion, four on-field officials and the VAR and AVAR, yet the decision sparks controversy among coaches, pundits, and spectators, reiterating how the laws of the game can be interpreted differently. Therefore, while the accuracy of decision making is important, the present study was more concerned with examining the external factors that may influence the decision-making process during the VAR protocol.

Since VAR was implemented in the English Premier League at the beginning of the 2019/2020 season, the Professional Game Match Officials Limited (PGMOL),

the body responsible for refereeing games in English professional football, have claimed to be consistently reviewing the use of VAR in the Premier League and often introduce changes at the start of each season. For example, at the beginning of the 2024/25 season, "referee's call" was introduced [13]. Following this, in-stadium announcements began at the start of the 2025/26 season where referees announce and explain their decisions at the pitch-side monitor over the PA system, in a bid to increase transparency with the crowd in attendance [14]. Also, for the 2025/26 season, the assistant referee will now remain on the touchline in line with the penalty spot when a penalty kick is taken. Prior to this, they would take a position on the goal-line (between one goalpost and the edge of the penalty area) to identify and alert the referee of goalkeeper encroachment. However, they are no longer required to do so, as the VAR is now solely responsible for identifying and alerting the referee if the goalkeeper has encroached during the penalty kick. While this final example is not explicitly relevant to the VAR protocol, it does highlight the growing role and responsibilities of the VAR/AVAR and how this is changing season-to-season making it necessary to assess the differences between decision-making across seasons.

One aspect of the VAR protocol that has remained constant is the location of the pitch-side monitor. As referees must remain visible during the review process to ensure transparency [11], it has meant that the pitch-side monitor is often positioned on one side of the pitch close to the half-way line. In most cases, this is close to the dugout of both the home and away teams and usually positioned in front of home fans, making the current research team curious as to whether home advantage influences the decisions at the pitch-side monitor.

Home advantage is the tendency for sporting teams to win more matches played at their home ground than those played away from home [15]. While this advantage may be due to the familiarity of the pitch, as well as dressing rooms, other facilities, and less travel, it could also be due to the support from a home crowd [16]. In fact, 60% of all competition points, across sports, are scored at home [17], as well as away teams receiving more yellow and red cards than home teams in football [18–21]. When investigating crowd effects, researchers have observed an increase in this apparent home team bias with increasing crowd size [19] and crowd density [21]. These findings are supported by evidence that home bias was reduced during the COVID-19 pandemic where attendance restrictions were enforced [22]. Sors et al. [22] reviewed all first and second division matches in England (Premier League/ Championship), Spain (La Liga/ Segunda División), Italy (Serie A/ Serie B), Germany (Bundesliga/ 2. Bundesliga), and France (Ligue 1/ Ligue 2) in the 2020/21 season, where restrictions remained in place and found that there were fewer home victories and more away victories than the previous five seasons where restrictions were not in place.

Kocsoy [23] also found that referees favour the home teams in terms of additional time awarded. When reviewing 9018 football matches across six European football leagues (England, France, Germany, Italy, Spain, Türkiye) between 2018 and 2023 they observed that 3550 of these matches went into additional time with a one-goal difference. A primary finding in this study was that referees kept the game forty-four seconds shorter when the home team were leading by a single goal compared to when the away team are leading by one goal at the end of the 90th minute. This figure is elevated to fifty seconds when in the presence of a crowd, and the more home fans in the stadium, the more likely the referee is to shorten the amount of additional time that is played. Thus, it appears that decision making can be influenced by the attendance in the stadium, as well as whether the decision is regarding the home or away team.

The time of the match the decision is taking place should also be considered as there is evidence that, like players, referees may also grow into a game. Mascarenhas et al. [24] assessed five FIFA-qualified referees across seven matches in the New Zealand Football Championship in the 2005/06 season. When analysing 144 incidents, it was found that referees made accurate decisions 64% of the time. However, this accuracy was lower in the first fifteen minutes of each half (51%) compared to all other parts of the match (70%). Although this is a relatively small sample in just one league, there is evidence that referees frequently make errors with correct decision rates ranging from 50% to 93.1% [25] of which Mascarenhas et al.'s [24] findings fall between. These findings, therefore, suggest that the time of the match can influence decision making and therefore could do so during the VAR protocol.

Decision making in football has historically and predominantly been the responsibility of the referee, however, now with the introduction of the VAR protocol, the decision making of the VAR, is now under scrutiny too. Recent research has found that age, experience, and status asymmetries between officials influence decision making [26]. When assessing VAR interventions in the Eredivisie, the Dutch top football league, between 2019 and 2021, Garretsen et al. [26] discovered that younger, less experienced, and lower-ranked VARs recommended more interventions and are more likely to have their recommendations dismissed by the on-field referee. These findings suggest that there are cognitive and social factors that contribute to decision making under uncertain, high-pressure situations and therefore, the current study was interested in examining the likelihood of VAR intervention across seasons, as well as whether attendance in the stadium played a role, given these factors are yet to be explored.

The current study, therefore, aimed to make three contributions to the literature. Firstly, to provide prevalence rates of maintain and overturn decisions at the pitch-side monitor between 2021–2025 in the English Premier League. Secondly, to explore the likelihood of referees maintaining their decision at the pitch-side monitor based on external factors such as the season in which the decision took place, whether the decision was regarding the home or away team, the scoreline at the time of the decision, the time in which the decision took place, and the attendance in the stadium. Finally, the current study also tested whether the season in which the decision took place, and stadium attendance influenced the likelihood of VAR intervention.

## Method

### Procedure

The research team created a Microsoft Excel document that consisted of English Premier League results from the first four seasons where the VAR protocol was implemented that were not interrupted by COVID-19 restrictions. This decision was made to amend for the potential confound of referees making decisions at the monitor in an empty stadium compared to one at full capacity. Therefore, the seasons that were reviewed were 2021/22, 2022/23, 2023/24 and 2024/25. As there are 380 matches in a Premier League season, the total number of matches reviewed was 1520. Reviews were conducted using the match reports on Sky Sports News' website. Details of these matches would then be recorded in the Microsoft Excel document. Reviews of these matches were conducted retrospectively, and during one current season as reviews commenced on 16/08/2024 and were completed on 26/05/2025.

There were three reasons that an on-field referee would be asked to review the pitch-side monitor: a goal, a red card, or a penalty. The original decision of the on-field referee was recorded as well as their decision after reviewing the pitch-side monitor. If the referee then changed their decision, this was recorded as an *overturn* and if they maintained their decision, this was recorded as a *maintain*. The time of the decision, the score at the time, and the attendance of the match was recorded to support our analysis of contextual factors. Two authors were responsible for the initial data extraction where six authors verified 20% of these inputs and concluded no errors were evident.

### Ethics

British Psychological Society ethical guidelines were adhered to throughout this study. Data extraction commenced following ethical approval from the University's Research Ethics Panel. As data extracted was secondary data, there was no participant information sheet, consent form, or debrief sheet required.

### Research questions

1. What are the odds that the on-field referee will overturn their original decision based on the season in which the decision took place, whether the decision was for the home team or away team, whether this team was winning, losing, or drawing at the time of the decision, the crowd attendance on the day, and the time of the decision?

2. What are the odds that the VAR will intervene based on the season in which the decision took place and the crowd attendance on the day?

## Aims

1. To calculate the prevalence of overturn and maintain decisions at the pitch-side monitor.

2. To calculate the odds that the referee would overturn their decision based on external factors.

3. To calculate the odds that the VAR would intervene based on external factors.

## Data analysis

Two Binary Logistic Regression tests were run to answer the two research questions. The first investigated the odds ratios of the independent variables (season, home advantage, scoreline, time of decision, and crowd attendance) on the likelihood of the on-field referee overturning their original decision. The second examined the odds ratios of the independent variables (season and crowd attendance) on the likelihood of the VAR intervening.

For the logistic regressions, season was categorised into one of the four seasons in which the decision took place. Home advantage was categorised by whether the team was playing at home or not, and scoreline was whether the team was winning, drawing, or losing at the time of the decision. Attendance at each match and the time of each decision was also recorded.

## Assumptions

As our outcome variables were categorical with two levels (overturn/maintain, intervened/did not intervene), we conducted binary logistic regressions. This analysis type requires little to no multicollinearity between independent variables, with Table 1 depicting this is the case for our data.

## Results

### Descriptive results

Data were collected from 1520 matches across four English Premier League seasons (2021/22, 2022/23, 2023/24 and 2024/25). The on-field referee was advised to consult the pitch-side monitor by the VAR on 250 occasions across 230 matches. When reviewing the pitch-side monitor, the on-field referee overturned their original decision 233 times (93.2%). Referees maintained their original decision 12 times (4.8%). There were 5 occasions (2%) where referees did not see the initial incident and therefore when reviewing the pitch-side monitor, were neither confirming nor denying their original

Table 1. Pearson correlations between independent variables.

|  | Season | Scoreline | Home Advantage | Time of decision | Match Attendance |
|---|---|---|---|---|---|
| Season | – | .077 | .003 | −.010 | .006 |
| Scoreline | .077 | – | .091 | .064 | .033 |
| Home Advantage | .003 | .091 | – | .060 | −.071 |
| Time of decision | −.010 | .064 | .060 | – | −.001 |
| Match Attendance | .006 | .033 | −.071 | −.001 | – |

\* - Correlation is significant at $p < .01$ (two-tailed)

decision, as they were viewing the incident for the first time. These five were subsequently removed from data analysis which amended the proportion of overturn decisions to 95.1% and maintained decisions to 4.9%. The number of times the referee was asked to the pitch-side monitor ranged from zero to two ($n$, 0 = 1290, 1 = 207, 2 = 19).

Of the 245 remaining decisions that were made, 148 (60.4%) were a decision that was directly related to the home team ($n$, Goal/ No Goal = 27 (18.4%), Red Card/ No Red Card = 24 (16.3%), Penalty/ No Penalty = 96 (65.3%)). There was one case where two decisions were to be made at the pitch-side monitor, where one influenced the other. In the match the original decision was no booking to the home team and no penalty to the away team which after review was overturned to a red card for the home team and penalty for the away team. As the initial decision was whether the home team had made a foul, and that this influenced whether a second decision regarding the penalty was required, this was deemed as one decision that was directly related to the home team. Ninety-seven (39.6%) decisions were directly related to the away team ($n$, Goal/ No Goal = 24 (24.7%), Red Card/ No Red Card = 22 (22.7%), Yellow Card/ No Yellow Card = 1 (1.1%), Penalty/ No Penalty = 50 (51.5%)).

Of the 233 (95.1%) overturned decisions, 139 (59.7%) were directly related to the home team ($n$, Goal/ No Goal = 24 (17.3%), Red Card/ No Red Card = 22 (15.8%), Penalty/ No Penalty = 93 (66.9%)), and 94 (40.3%) were directly related to away team ($n$, Goal/ No Goal = 22 (23.4%), Red Card/ No Red Card = 22 (23.4%), Yellow Card/ No Yellow Card = 1 (1.1%), Penalty/ No Penalty = 49 (52.1%)).

Of the 12 (4.9%) maintain decisions, nine (75%) were directly related to the home team ($n$, Goal/ No Goal = 3 (33.3%), Red Card/ No Red Card = 3 (33.3%), Penalty/ No Penalty = 3 (33.3%)), and three (25%) were directly related to the away team ($n$, Goal/ No Goal = 2 (66.6%), Penalty/ No Penalty = 1 (33.3%)).

It was most common for the VAR to intervene while the match was level ($n$ = 115, 46.9%) as opposed to intervening for an incident involving the winning ($n$ = 47, 19.2%) or losing team ($n$ = 83, 33.9%). Likewise, it was more common for the VAR to intervene later in the match ($n$, First quarter = 28 (11.4%), Second quarter = 65 (26.5%), Third quarter = 62 (25.3%), Final quarter = 90 (36.8%), *Minutes*, $M$ = 55.59, $SD$ = 24.61, $Range$ = 4–90). Attendances across the four seasons ranged from 9972 to 75546 ($M$ = 39612.18, $SD$ = 16607.70).

## Binary logistic regression

The initial Binary Logistic Regression was performed to assess the impact of a set of predictor variables on the odds that the referee would overturn their original decision. The first model contained five independent variables (season, home advantage, scoreline, time of decision, and crowd attendance). The full model containing all predictors was not statistically significant $\chi^2$ (8, $N$ = 245) = 9.946, $p$ = .269, indicating that the model was unable to distinguish between the referee's decision to overturn or maintain their original decision. The model as a whole correctly classified 95.1% of cases. As shown in Table 2, none of the independent variables made a unique statistically significant contribution to the model.

A second Binary Logistic Regression was performed to assess the impact of a set of predictor variables on the odds that the VAR would intervene. The model contained two independent variables (season and crowd attendance). The full model containing both predictors, again, was not statistically significant $\chi^2$ (4, $N$ = 1535) = 2.424, $p$ =.658, indicating that this model was also unable to distinguish between whether the VAR intervened or not. The model as a whole correctly classified 84% of cases. As shown in Table 3, neither of the independent variables made a unique statistically significant contribution to the model.

## Maintain decisions

Due to the exploratory nature of this study, and the gross imbalance of overturn and maintain decisions, we deemed it necessary to describe the maintain decisions. With there being so few occasions where on-field referees maintained their decisions, the reader is provided with descriptive statistics regarding these events.

**Table 2. Logistic Regression predicting the likelihood of the referee maintaining their original decision at the pitch-side monitor.**

| | B | SE | Wald | df | p | Odds ratio | 95% CI for Odds Ratio | |
|---|---|---|---|---|---|---|---|---|
| | | | | | | | Lower | Upper |
| Attendance | .000 | .000 | 1.515 | 1 | .218 | 1.000 | 1.000 | 1.000 |
| Time | .010 | .013 | .658 | 1 | .417 | 1.010 | .985 | 1.036 |
| Season 21/22 | −.104 | 1.435 | .005 | 1 | .942 | .901 | .054 | 15.009 |
| Season 22/23 | 1.858 | 1.108 | 2.812 | 1 | .094 | 6.414 | .731 | 56.292 |
| Season 23/24 | 1.142 | 1.140 | 1.003 | 1 | .316 | 3.134 | .335 | 29.289 |
| Scoreline – Winning | −.037 | .911 | .002 | 1 | .968 | .964 | .162 | 5.742 |
| Scoreline – Drawing | .164 | .699 | .055 | 1 | .815 | 1.178 | .299 | 4.641 |
| Home Team | .900 | .706 | 1.624 | 1 | .203 | 2.459 | .616 | 9.811 |
| Constant | −4.389 | 1.588 | 7.637 | 1 | .006 | .012 | | |

**Table 3. Logistic Regression predicting the likelihood of the VAR intervening.**

| | B | SE | Wald | df | p | Odds ratio | 95% CI for Odds Ratio | |
|---|---|---|---|---|---|---|---|---|
| | | | | | | | Lower | Upper |
| Attendance | .000 | .000 | .278 | 1 | .598 | 1.000 | 1.000 | 1.000 |
| Season 21/22 | −.219 | .202 | 1.174 | 1 | .279 | .804 | .541 | 1.194 |
| Season 22/23 | −.128 | .204 | .390 | 1 | .532 | .880 | .590 | 1.313 |
| Season 23/24 | −.278 | .200 | 1.944 | 1 | .163 | .757 | .512 | 1.120 |
| Constant | 1.909 | .226 | 71.291 | 1 | .001 | 6.749 | | |

Of the twelve maintain decisions over the four seasons analysed, nine (75%) were regarding the home team and eight (67%) came during the second half (Range in minutes, 21–90, $M = 61.50$, $SD = 25.22$). Half of these decisions were made when the game was balanced with four (33%) regarding the losing team and two (17%) regarding the winning team. There were ten different referees involved in these twelve decisions, with two maintaining their original decision twice in front of crowds ranging from 10,126–73,711 ($M = 34618$, $SD = 17494.80$). There were eight different VARs with two intervening twice and one intervening three times. There were twelve different AVARs for each maintain decision. Five of the decisions were regarding a goal, four a penalty, and three a red card.

## Discussion

The aims of this study were i) to calculate the prevalence of overturn and maintain decisions at the pitch-side monitor, ii) to calculate the odds that the referee would overturn their decision based on external factors, iii) to calculate the odds that the VAR would intervene based on external factors. This study has evidenced that referees in the English Premier League overturned their original decision over 95% of the time across the four-season period and that referees are consistent in these decisions across seasons. It was also found that home advantage, scoreline, time of decision, and crowd attendance do not sway this decision, and that season and attendance did not influence whether the VAR would intervene.

Referees overturning their original decision in 95% of cases is high but may be expected, as the reason they are being advised to consult the pitch-side monitor is due to a potential error. However, prior to this study, the prevalence of overturn decisions was unknown, and therefore this gap is filled. Future researchers are better informed when developing their hypotheses regarding this area with the knowledge that referees change their original decision nineteen in twenty times.

Without this knowledge, the current research team utilised an exploratory approach to investigate the prevalence of overturn and maintain decisions at the pitch-side monitor. The research team decided to explore the impact of potential variables that have been found to influence referee decisions in the past such as home advantage [15,23], scoreline,

the time of the match [24], and match attendance [19,21], and due to the ever-changing nature of the VAR protocol, we assessed whether season influenced the referee's decision. Season and match attendance were also investigated to determine whether they influenced if the VAR would intervene or not. As there was such a discrepancy in classification of our dependent variables i) 5% maintain decisions v. 95% overturn decisions, ii) 15% VAR intervention v. 85% no VAR intervention, the binary logistic regressions were rendered less useful in determining the likelihood of the referee maintaining their decision or the VAR intervening based on the external factors we were interested in. This finding provides support for PGMOL by suggesting that their officials, both on-field and VMOs are consistent in their decisions and are not influenced by external factors that have in the past been reported to affect decision making.

From this, the low prevalence of maintain decisions may be the more interesting aspect of these findings. For example, three quarters of maintain decisions were regarding the home team, and while this is a relatively small sample (9 of 12), this finding could be explained by referees implicitly favouring the home team as has been reported in the past [15,23]. As the sample of this occurrence is very small, it could simply be explained by coincidence. However, the location of the pitch-side monitor must also be considered. As referees must remain visible during the review process [11], it leaves few options for the location of the pitch-side monitor. Four enclosed stands comprise all 25 stadia included in the current study, which means the pitch-side monitor will always be set up near a set of supporters. Invariably, the position is close to the dugout of both the home and away teams which is often just yards away from home supporters, which could invoke some proximity bias in the on-field referee. Proximity bias is the tendency to favour those that are closer to us, with a popular example being office managers favouring those working in the office over those working remotely [27]. In the current context, it is possible that the home supporters and dugout have an influence on the rate of maintain decisions regarding the home team simply because they are closer to the referee. Likewise, decisions that regard the away team may be easier to overturn when the supporters are not close to the referee as they make their decision at the pitch-side monitor. This proximity bias may also explain why home bias reduced during COVID-19 restrictions where there was no crowd to influence the referee [22], whether implicitly or explicitly.

## Practical implications

Based on these findings, IFAB may wish to reconsider their 8th principle of the VAR protocol [11] that states "The referee must remain 'visible' during the review process to ensure transparency." The inverted commas surrounding the word *visible* leaves the method of this open to interpretation. Our interpretation of this is that referees are permitted to be visible in a range of modalities, and therefore IFAB perhaps do not need to review this principle, but instead, the English Premier League and the PGMOL may wish to review their interpretation of this principle. From the findings presented in the current study, we see that there is a tendency for referees to maintain their decision when that decision concerns the home team. The pitch-side monitor is often located yards away from home team fans and given our knowledge on proximity bias [22,27] moving the pitch-side monitor to inside the tunnel may reduce this bias and increase decision accuracy. If the decision is still recorded, and televised for supporters in the stadium, then the 8th principle of the VAR protocol set by IFAB [11] is still satisfied.

## Limitations

The findings of the present study have merit by presenting the prevalence of overturn/ maintain decisions at the pitch-side monitor in the Premier League over the last four seasons. Our plan of analysis to investigate the external factors that may influence the referee at the pitch-side monitor was also based on evidence [15,16,18–24]. However, due to the large imbalance between overturn and maintain decisions, the ability to perform binary logistic regression appropriately was hindered. With that said, two researchers were responsible for the initial reviewing of matches while six of the team member-checked 20% of the sample of matches. The research team are therefore confident that the dataset is accurate and that the high prevalence of overturned decisions is valid. From this, researchers' future hypotheses on

this topic will be embedded within these prevalence rates, and appropriate statistical analyses can be proposed prior to data collection.

The reader must also be reminded that much of the discussion is speculation of the psychology of match officials in football. At present there are no data regarding the thoughts and feelings of match officials before, during, and after the current VAR protocol, which is why a speculative approach to explaining the high prevalence of overturn decisions has been adopted. Future studies should aim to interview match officials to uncover their psychological processes during the current VAR protocol and gather their opinions on how it can be improved.

Moreover, there are two reasons that a review can occur; i) the VAR (or another match official) recommends a review, ii) the referee suspects that something serious has been missed [28]. The secondary data collection method using Sky Sports match reports did not report on whether reviews at the monitor were due to the VAR recommending a review or whether the referee had called for the review themselves. This therefore could limit our data as our data assumes that the VAR recommended each review. Although, we believe this is a reasonable assumption, and that most if not all reviews were recommended by the VAR, the reader should keep this in mind when interpreting these findings.

## Conclusion and future directions

To conclude, the present study aimed to i) calculate the prevalence of overturn and maintain decisions at the pitch-side monitor, ii) calculate the odds that the referee would overturn their decision based on external factors, iii) calculate the odds that the VAR would intervene based on external factors. The current study was successful in achieving these aims as data analysis revealed a prevalence of 95% overturned decisions in the English Premier League between 2021/22 and 2024/25. It was also found season, home advantage, scoreline, time of decision, and crowd attendance, do not influence this decision. VAR/AVAR intervention was also not influenced by season or crowd attendance. Due to the large imbalance between overturn and maintain decision, the research team have discussed some of the potential reasons for rarer occurrence of referees maintaining their decision at the pitch-side monitor, such as proximity bias. Practical implications based on these findings could include referees reviewing their decision at a monitor away from the pitch, but with this still recorded for transparency purposes. Following on from this work, it would also be useful to conduct qualitative interviews with English Premier League officials to investigate their thoughts and feelings towards external factors that may or may not influence decision making associated with the current VAR protocol.

## Author contributions

**Conceptualization:** Daniel Walker, Jade L. Jukes.

**Data curation:** Daniel Walker, Cara E. Staniforth, Jordan Thomas, Fallon Parker, Usman Khizar, Nathaniel J. Lawton, Tamryn Vermaak, Jade L. Jukes.

**Formal analysis:** Daniel Walker, Cara E. Staniforth, Jordan Thomas, Fallon Parker, Usman Khizar, Nathaniel J. Lawton, Tamryn Vermaak, Jade L. Jukes.

**Investigation:** Daniel Walker.

**Methodology:** Daniel Walker, Cara E. Staniforth, Jade L. Jukes.

**Project administration:** Daniel Walker.

**Resources:** Daniel Walker.

**Software:** Daniel Walker.

**Supervision:** Daniel Walker.

**Validation:** Daniel Walker.

**Visualization:** Daniel Walker.

**Writing – original draft:** Daniel Walker.

**Writing – review & editing:** Daniel Walker, Cara E. Staniforth, Jade L. Jukes.

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
