## [Decision Letter · Decision Letter 0]

4 Feb 2026

Dear Dr. Walker,

Thank you for submitting your manuscript to PLOS ONE. After careful consideration, we feel that it has merit but does not fully meet PLOS ONE’s publication criteria as it currently stands. Therefore, we invite you to submit a revised version of the manuscript that addresses the points raised during the review process.

We look forward to receiving your revised manuscript.

Kind regards,

Additional Editor

PLOS One

Journal Requirements:

Reviewers' comments:

Reviewer's Responses to Questions

**Comments to the Author**

1. Is the manuscript technically sound, and do the data support the conclusions?

Reviewer #1: Yes

Reviewer #2: Yes

2. Has the statistical analysis been performed appropriately and rigorously?

Reviewer #1: Yes

Reviewer #2: Yes

3. Have the authors made all data underlying the findings in their manuscript fully available?

Reviewer #1: Yes

Reviewer #2: Yes

4. Is the manuscript presented in an intelligible fashion and written in standard English?

Reviewer #1: Yes

Reviewer #2: Yes

Reviewer #1: The results were thoroughly analyzed, and the data interpretation was performed accurately and cautiously.

The additional details (following the initial review) are adequately meet expectations.

Furthermore, this article demonstrates sufficient scientific rigor for publication in the journal.

In conclusion, this article demonstrates sound scientific quality and is suitable for publication as is.

Reviewer #2: Thank you for writing this manuscript. It is indeed very beneficial for research in the area of VAR in football, particularly the English Premier League (EPL). I do have some questions that I would like further clarification:

Line 73:

How is match-changing incidents conceptually defined?

Line 116-130:

Are any of these research referring specifically to the English Premier League? It does not indicate which league, only a general mention of the countries. Since this study is specific to the (EPL), you might want to specify which studies are referring to the EPL.

Line 237-259:

You might want to add the percentage (%) as well for the description of statistics to make it more reader friendly. For example:

Of the 245 remaining decisions that were made, 148 (60.4%) were a decision that was directly

238 related to the home team.

Future Directions:

What do you think about qualitative research, interviewing the experience of referees themselves, of whether they think external factors might influence their decisions? This comes into light after Mike Dean admitted on national television that his decision was influenced by being "mates" with other referees.

**Do you want your identity to be public for this peer review?** For information about this choice, including consent withdrawal, please see our Privacy Policy

Reviewer #1: No

Reviewer #2: No

---

## [Author Response · Author response to Decision Letter 1]

23 Feb 2026

Please refer to response to reviewers document attached.

---

## [Decision Letter · Decision Letter 1]

9 Mar 2026

Psychology at the screen: Investigating the current VAR protocol

PONE-D-25-60174R1

Dear Dr. Walker,

We’re pleased to inform you that your manuscript has been judged scientifically suitable for publication and will be formally accepted for publication once it meets all outstanding technical requirements.

Kind regards,

Wanli Zang, Ph.D.

Guest Editor

PLOS One

Additional Editor Comments (optional):

Reviewers' comments:

Reviewer's Responses to Questions

**Comments to the Author**

Reviewer #2: All comments have been addressed

2. Is the manuscript technically sound, and do the data support the conclusions?

Reviewer #2: Yes

3. Has the statistical analysis been performed appropriately and rigorously?

Reviewer #2: Yes

4. Have the authors made all data underlying the findings in their manuscript fully available?

Reviewer #2: Yes

5. Is the manuscript presented in an intelligible fashion and written in standard English?

Reviewer #2: Yes

Reviewer #2: (No Response)

**Do you want your identity to be public for this peer review?** For information about this choice, including consent withdrawal, please see our Privacy Policy

Reviewer #2: No

---

## [Editor Report · Acceptance letter]

PONE-D-25-60174R1

PLOS One

Dear Dr. Walker,

I'm pleased to inform you that your manuscript has been deemed suitable for publication in PLOS One. Congratulations! Your manuscript is now being handed over to our production team.

Kind regards,

on behalf of

Dr. Wanli Zang

Guest Editor

PLOS One